# Giant Parathyroid Adenoma of the Posterior Mediastinum

**DOI:** 10.3390/medicina60101666

**Published:** 2024-10-11

**Authors:** Rokas Jagminas, Jolanta Jocienė, Vygantė Maskoliūnaitė, Žymantas Jagelavičius

**Affiliations:** 1Faculty of Medicine, Vilnius University, LT-03225 Vilnius, Lithuania; 2Clinic of Chest Disease, Immunology and Allergology, Institute of Clinical Medicine, Faculty of Medicine, Vilnius University, LT-03225 Vilnius, Lithuania; jolanta.jociene@santa.lt (J.J.); zymantas.jagelavicius@santa.lt (Ž.J.); 3Department of Pathology and Forensic Medicine, Institute of Biomedical Sciences, Faculty of Medicine, Vilnius University, LT-03225 Vilnius, Lithuania; vygante.maskoliunaite@vpc.lt

**Keywords:** posterior mediastinum, thoracic surgery, parathyroid adenoma, VATS, RATS, neuroendocrine tumour

## Abstract

Ectopic parathyroid adenoma is a rare pathology. We present a clinical case of a giant ectopic parathyroid adenoma (PA) in an unusual location, which brought significant diagnostical and therapeutical challenges. The tumour in the mediastinum was found incidentally on chest computed tomography (CT). A biopsy was conducted, and histological examination revealed a well-differentiated neuroendocrine tumour. The tumour was excised via right thoracotomy. The final histological examination revealed a parathyroid adenoma, which was unexpected for our team. After three years of observation, there is no evidence of tumour recurrence.

## 1. Introduction

Ectopic inferior parathyroid glands are often found in the anterior mediastinum in association with the thymus or the thyroid gland. The most common localisation for ectopic superior parathyroids is the tracheoesophageal groove and retroesophageal region. The prevalence of ectopic parathyroid location is 15.9%, with 11.6% discovered in the neck and 4.3% in the mediastinum. We present a case of a giant ectopic parathyroid adenoma in a highly unusual location, leading to diagnostic and treatment challenges.

## 2. Case Presentation

A 70-year-old patient was hospitalized after falling that caused mild head trauma. She was hemodynamically stable. The laboratory tests showed increased pro-inflammatory markers. Anterior and lateral chest roentgenography was conducted and a non-homogenic consolidation was observed for the first time. Treatment with antibiotics was prescribed. There were no signs of other organ system damage. Furthermore, a computed tomography scan revealed alterations in the posterior mediastinum (Figure 1). A tumour measuring up to 5.5 cm in its longest dimension, with contrast accumulation, was found, extending from the second to the fourth thoracic vertebra and situated near the posterior wall of the oesophagus.

The patient had primary (essential) hypertension and her BMI was 28.87 kg/m^2^. Her arterial hypertension was well controlled with nebivolol and zofenopril. There were no allergies to medications. There was no gastritis, duodenitis with ulcers, osteopenia, or osteoporosis in her medical history that might be related to hyperparathyroidism.

The patient was transmitted to the thoracic surgery department for further investigation of the tumour in the posterior mediastinum. Since there were no complaints or symptoms related to hyperparathyroidism, no laboratory tests were performed. She had no signs of choking. The oesophageal mucosa was normal, but after esophagogastroduodenoscopy, upper-third oesophagus compression was seen. A transoesophageal endoscopic ultrasound-guided fine-needle aspiration biopsy was conducted, leading to a diagnosis of a well-differentiated neuroendocrine tumour based on histological findings. However, immunohistochemical testing was limited due to the small amount of tumour tissue obtained from the aspiration biopsy. The imaging supported the presence of a solitary tumour, which could potentially have originated from the oesophagus or thymus.

The patient then underwent surgery, during which a 5.5 × 4.2 × 3.1 cm, 52 g cystic tumour localised in the posterior mediastinum was removed via right thoracotomy (Figure 2). Additionally, three fragments of histologically verified normal parathyroid tissue and one lymph node with reactive lymphadenopathy were removed.

Microscopically, a nodular tumour encapsulated by a fibrous capsule was identified in the main fragment (Figure 3). The tumour nodules were separated with fibrous septa. Different histological patterns were identified: predominant solid zones with trabeculae, less common variously sized follicles, acini, and cystic spaces. The tumour cells had a moderate to abundant amount of clear, focally eosinophilic cytoplasm with distinct borders and small, round, centrally located nuclei. Mitotic figures were scant: approximately two mitoses per 50 high-power fields (HPFs). Few cystic spaces were filled with haemorrhages or haemosiderophages. Also, there were fibrotic areas, most likely associated with a previous biopsy. At the tumour periphery, small compressed normal parathyroid tissue inclusions were identified. There were no histological signs of malignancy, including prominent cytological atypia or tumour necrosis, infiltrative growth, or any evidence of invasion into surrounding tissues or perineural or lymphovascular spaces.

Immunohistochemically, the tumour was positive for parathyroid hormone (PTH), chromogranin A, PAX8, and GATA3. Staining for synaptophysin, thyroglobulin, and TTF1 was negative. Reactions for p53, p27, and Bcl-2 were heterogeneous. The Ki67 proliferative index was low (the full immune profile is presented in Table 1). The final pathological diagnosis was a parathyroid adenoma in the ectopic parathyroid tissue from the posterior mediastinum.

Postoperatively, no signs of hyperparathyroidism were observed, and the recovery was uneventful. Genetic testing was also performed, and there were no pathological findings, including MEN1 syndrome. In addition, a bone density scan (DXA) was also performed, and the T measurement index was –1.35 (osteoporosis is identified when T < −2.5). There were no signs of nephrolithiasis. The patient had a post-operative oncologist consultation, and she is under continued surveillance up by an endocrinologist (laboratory results are presented in Table 2) and thoracic surgery team, with evidence of recurrence three years after surgery.

## 3. Discussion

The parathyroid glands, which are part of the neuroendocrine system, are typically positioned behind the thyroid glands, but in front of other neck structures. Their main function is to regulate calcium homeostasis. In book-based anatomy, the inferior parathyroid glands are found at the lower poles of the thyroid gland, while the superior parathyroid glands are found near the upper poles [1]. Although most individuals have four parathyroid glands, the total number can vary from one to as many as twelve [2]. The superior parathyroid glands descend from the fourth pharyngeal pouch within the thyroid, while the inferior parathyroid glands migrate alongside the thymus from the third pharyngeal pouch. As a result of their migratory path extending from the mandibular angle to the mediastinum, ectopic glands can be situated anywhere along this central line. The rarest locations for ectopic parathyroid glands include within the n. vagus or pharynx and behind the oesophagus [3].

In our case, there was a discussion about what type of tissue mass from the posterior mediastinum this was. Statistically, the majority of tumours are neuroendocrine tumours, lymphomas, benign and malignant soft-tissue tumours, oesophageal tumours, and metastasis, respectively. Although there was a possibility of performing a tumour excision without a biopsy to avoid tumour seeding, histological tumour-type confirmation was essential in the differential diagnosis. For example, lymphomas, a relatively common type of tumour in the posterior mediastinum, are typically treated with systemic therapies such as chemotherapy, immunotherapy, or targeted therapy and generally do not require surgical intervention for treatment. That is why we performed a transoesophageal endoscopic ultrasound-guided fine-needle aspiration biopsy. The pathologist subsequently confirmed the neuroendocrine cell origin of this tumour, leading to the possible diagnosis of a primary oesophageal or thymic well-differentiated neuroendocrine tumour (NET G1 or G2). Surprisingly, the pathological examination of the surgical specimen revealed that the tumour originated from ectopic parathyroid gland tissue in the posterior mediastinum. The final diagnosis of parathyroid adenoma was established following a comprehensive evaluation, during which other types of neuroendocrine tumours were excluded based on typical histological and immunohistochemical findings. Additionally, there were no histological signs of malignancy that would support a diagnosis of parathyroid carcinoma.

Our patient’s mass in the posterior mediastinum was giant and the localisation of the tumour was not favourable for video-assisted thoracoscopic surgery (VATS). For patient safety and better outcomes, we decided to perform a right thoracotomy. Thoracotomy has been used for surgical excision of mediastinal PAs that are difficult to excise using the cervical approach. This method in our case had the advantages of accuracy in tumour identification and good operative view.

Unfortunately, there is no “gold standard” surgical method for ectopic parathyroid adenomas yet. Any neuroendocrine tumours in the posterior mediastinum that are not removable through the neck excision are usually approached by sternotomy or anterior thoracotomy. Posterior mediastinal PAs usually need posterolateral thoracotomy. It is important to mention that any thoracic approaches demand considerable anaesthetic and surgical knowledge [4].

Conventional approaches for mediastinal EPA are sternotomy or thoracotomy. These approaches have been largely associated with significant complications as well as phrenic and recurrent laryngeal nerve injuries, innominate vein laceration, mediastinitis, and death [5]. Sternotomy and thoracotomy, which were standard approaches in the past, are associated with increased morbidity. Additionally, other studies have reported more complications and prolonged hospital stays exceeding 10 days [5,6]. Our patient was hospitalized for 7 days. In contrast, parathyroidectomy using video-assisted thoracoscopic surgery (VATS) and robotic-assisted thoracoscopic surgery (RATS) offers several advantages, including reduced postoperative pain, fewer complications, shorter hospitalisation, quicker return to daily activities, and improved cosmetic outcomes.

It is important to highlight that a mediastinal tumour in this case was a completely incidental finding. There was no consideration of measuring PTH or calcium levels, because the patient was completely asymptomatic. Only after the complete removal of the mass did we conclude that it was a giant parathyroid neuroendocrine tumour, with immunohistochemical analysis confirming PTH expression in the surgical specimen. According to the present literature, most ectopic parathyroid glands stay asymptomatic, without clinical manifestations [7]. Furthermore, it is noticed that giant parathyroid adenomas could have a lower incidence of symptoms despite increased PTH production [8]. Consequently, surgery is generally not recommended unless the patient exhibits symptoms.

Another recognised cause of primary hyperparathyroidism is an autosomal-dominant condition known as multiple endocrine neoplasia type 1 (MEN1). The algorithm-based treatment for MEN1-associated primary hyperparathyroidism is surgical removal, though this becomes more complex when the parathyroid gland is ectopic. Ectopic parathyroid glands are more often observed in MEN1 patients than in those with sporadic hyperparathyroidism. The overall prevalence and incidence of MEN1 remain uncertain, though its prevalence is estimated to be approximately 1 in 10,000 to 1 in 30,000 [9].

Giant PAs are defined as those with a weight of more than 3.5 grams. The median weight of PAs in that study was 0.61 grams, with a range between 0.05 and 29.93 grams [8]. In our case, the tumour weighed 52 grams, was encapsulated, and contained dark-brown liquid inside it. The diameter of the tumour was 5.5 × 4.2 × 3.1 cm. It also consisted of an oval piece of rubbery soft red and tan tissue, which histologically corresponded with post-interventional changes as fibrosis and areas of haemorrhages.

It is necessary to mention that it is usually difficult to measure the outcomes of parathyroid neuroendocrine tumours. Currently, there is extraordinarily little literature available outlining its clinical aspects, diagnostic modalities, optimum treatment options, and prognostic markers regarding survival rates of parathyroid neuroendocrine tumours. In contrast, parathyroid carcinoma previously reported 5-year and 10-year survival rates are 76–85% and 49–77%, respectively [10]. Our patient had no signs of tumour relapse after three years of observation.

## 4. Conclusions

A giant ectopic parathyroid adenoma in the posterior mediastinum is an extremely rare case. Safe and radical removal of the tumour is essential for successful treatment, but quite often the final pathological diagnosis is determined only after surgery. Asymptomatic parathyroid adenomas found deeper in the mediastinum can be a challenge to diagnosis and surgery. It is important to mention that a parathyroid adenoma should be suspected in cases of any mass in the mediastinum.

## Figures and Tables

**Figure 1 medicina-60-01666-f001:**
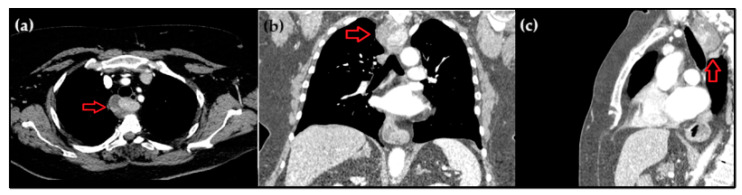
Axial (**a**), coronal (**b**), and sagittal (**c**) CT of the chest with contrast, showing a mass in the posterior mediastinum (red arrow).

**Figure 2 medicina-60-01666-f002:**
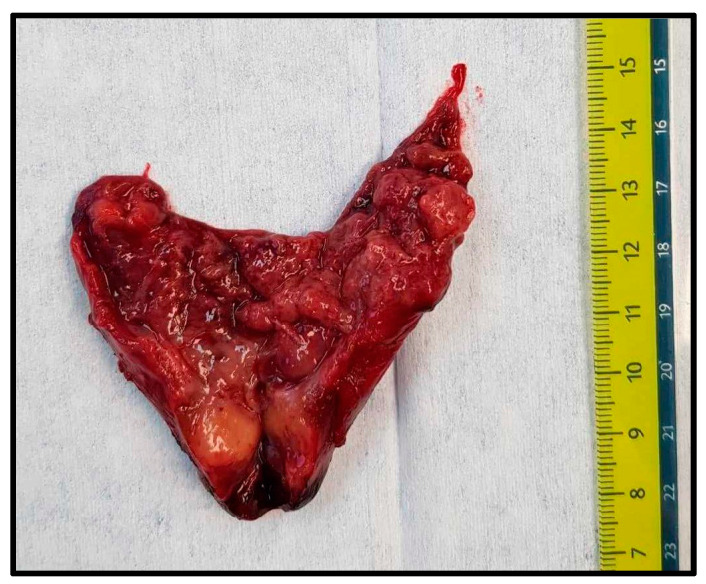
Removed tumour.

**Figure 3 medicina-60-01666-f003:**
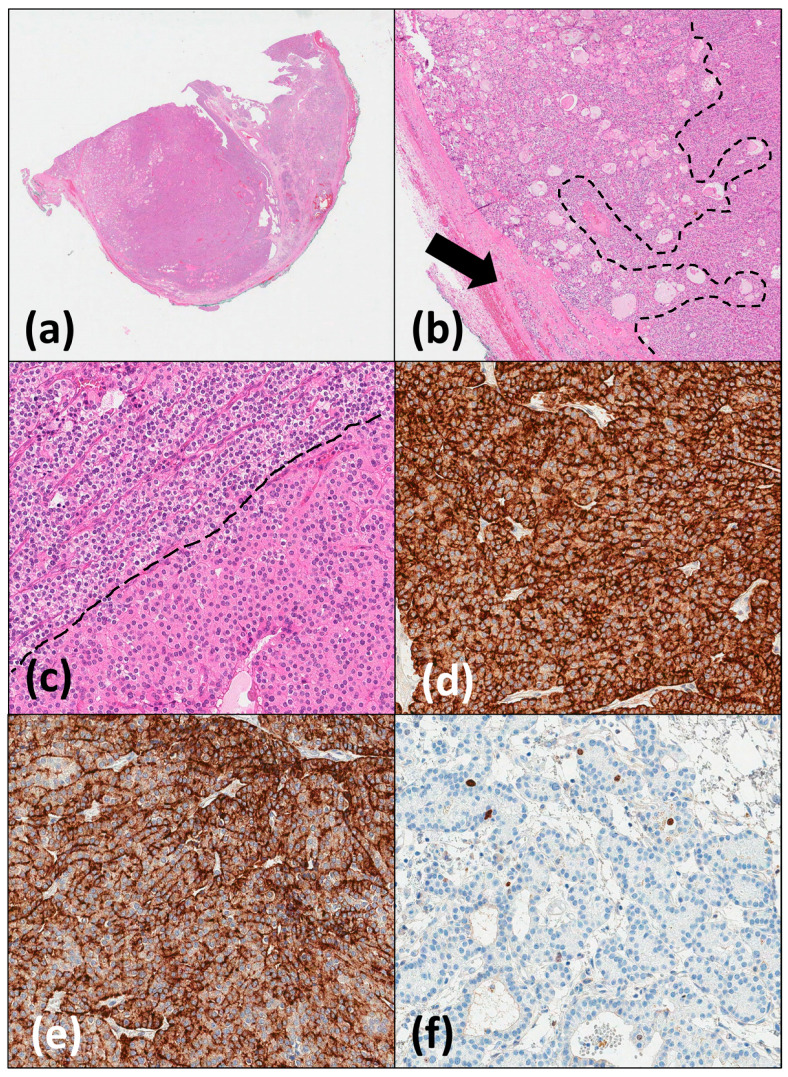
Microscopic slides from resected ectopic parathyroid adenoma: (**a**) panoramic view of encapsulated tumour, haematoxylin and eosin (HE), magnification ×5; (**b**) cystic (left) and solid trabecular (right) tumour structures with thick fibrotic capsule and compressed normal parathyroid tissue (arrow), HE, magnification ×40; (**c**) predominant chief (top left) and focal oxyphilic/oncocytic (bottom right) tumour cells, HE, magnification ×200; (**d**) strong positive immunohistochemical parathormone reaction in the cytoplasm, PTH, magnification ×200; (**e**) variable positive immunohistochemical reaction for chromogranin A, magnification ×200; (**f**) low Ki67 proliferative index in the tumour cells, magnification ×200.

**Table 1 medicina-60-01666-t001:** Comparison of immunohistochemical tumour profile with reaction intensity and percentage of positive tumour cells in the biopsy and the surgical specimen. Abbreviations: ND—not done; (−)—negative; (+)—weak positivity; (++)—moderate positivity; (+++)—strong positivity.

Immunohistochemical Marker	Biopsy	Surgical Specimen
Cytokeratins AE1/AE3	(+++) 100%	ND
Chromogranin A	(++/+++) 100%	(+/+++) 80%
Synaptophysin	(−)	(-)
PTH	ND	(+++) 95%
TTF1	(−)	(−)
Thyroglobulin	ND	(−)
GATA3	ND	(+/+++) 80%
PAX8	ND	(+/+++) 100%
CDX2	(−)	ND
Serotonin	(−)	ND
p53	ND	(+/++) 5%
p27	ND	(+/+++) 25%
Bcl-2	ND	(+/+++) 20%
Ki67	(++/+++) < 2%	(++/+++)~2% (in “hot spots” < 5%)

**Table 2 medicina-60-01666-t002:** Laboratory work-up through the treatment. RV—reference value used in Vilnius University Hospital Santaros Clinics (VUL SK) laboratory, Vilnius, Lithuania.

Laboratory Measurement	1 Month after Surgery	Follow-Up after 9 Months	Follow-Up after 18 Months	Follow-Up after 24 Months
Ionised calcium mmol/L (RV: 1.00–1.30 mmol/L)	1.28	1.24	1.33	1.15
Total calcium mmol/L (RV: 2.12–2.62 mmol/L)	2.28	2.31	2.58	2.20
Total phosphorus mmol/L (RV: 0.80–1.45 mmol/L)	-	0.94	0.96	0.99
Parathyroid hormone pmol/L (RV: 1.60–6.90 pmol/L)	49.95	42.99	25.15	22.89
25-OH vitamin D nmol/L (RV: 75.00 to 125.00 nmol/L)	27.6	34.1	79.6	60.3
eGFG mL/min/1.73 m^2^	59	58	53	52

## Data Availability

The data presented in this study are available on request from the corresponding author.

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
