# Peer review of "Giant Parathyroid Adenoma of the Posterior Mediastinum"

_medicina, 2024, doi:10.3390/medicina60101666_

Round 1

Reviewer 1 Report

Comments and Suggestions for Authors

Dear Editor and Authors,

The authors presented a clinical report of a 70-year-old woman with a giant parathyroid adenoma incidentally found in the posterior mediastinum. The case represents a rare case in the clinic. Therefore, the presence of such a case example in the literature should lead clinicians to consider the possibility that parathyroid adenomas may also be located in the posterior mediastinum. However, I believe there are some deficiencies in the case presentation, and correcting these may increase the reader's interest in the case. I suggest some changes that I think are necessary below.

1. The English language should be revised thoroughly to facilitate reader understanding and ensure fluency. The paper contains numerous typo and grammatical errors.

2. The keywords should be rewritten. There is no need to write both ectopic parathyroid adenoma and parathyroid adenoma. Prefer the former.

3. The paper presents the case of a 70-year-old woman. But we know nothing about her. Did the woman have other diseases? Was she taking medication? What information was included in the patient's history and family history?

4. It is stated that the patient underwent a CT scan by chance. Is this a common situation? Why was a CT scan performed on someone who had no symptoms? Is this a routine practice at your center? Did the patient really have no symptoms?

5. We do not know the patient's laboratory values ​​at the time of diagnosis. Were serum calcium, PTH, etc. values ​​really normal? Perhaps she was receiving calcium-lowering medical treatment (such as cinacalcet). Provide a table with the patient's laboratory values ​​at the time of diagnosis (such as PTH, calcium, phosphorus, albumin, vitamin D, creatine, urine calcium level, etc.). Also, did she have osteoporosis or nephrolithiasis?

6. Although it is a very rare condition, such a mass in the posterior mediastinum could also be a pheochromocytoma or paraganglioma. I would definitely consider checking the catecholamine levels. In this way, I would have had the chance to prepare before surgery and would not have put the patient at risk. Have you considered checking the catecholamines? (To prevent a possible pheochromocytoma crisis.).

7. Considering the possibility of seeding the tumor, I would not consider performing a biopsy on such a large mass in such a localization. I would consider performing surgery directly. What do you think about this?

8. Please include the images of the histopathological examination (light microscopy imaging). Present which stains were performed histopathologically and which were positive and which were negative, in a tabular format.

9. Finally, I think the diagnosis should definitely be confirmed by another pathologist.

I wish the authors success.

Comments on the Quality of English Language

The English language should be revised thoroughly to facilitate reader understanding and ensure fluency. The paper contains numerous typo and grammatical errors.

Author Response

  1. The English language should be revised thoroughly to facilitate reader understanding and ensure fluency. The paper contains numerous typo and grammatical errors.

Thank you for your comment. We will send our paper to a native British-English speaker to check for grammatical errors.

  1. The keywords should be rewritten. There is no need to write both ectopic parathyroid adenoma and parathyroid adenoma. Prefer the former.

We agree with your point. We made changes to the keywords.

  1. The paper presents the case of a 70-year-old woman. But we know nothing about her. Did the woman have other diseases? Was she taking medication? What information was included in the patient's history and family history?

We agree with your comment. At first, we wanted to exclude all the information that is not related to her case to avoid readers' confusion and distraction. You will see our updated report in an attached file. However, I am adding additional information here as well.

  1. It is stated that the patient underwent a CT scan by chance. Is this a common situation? Why was a CT scan performed on someone who had no symptoms? Is this a routine practice at your center? Did the patient really have no symptoms?

70-year-old patient was hospitalized after falling that caused mild head trauma. She was hemodynamically stable, her pulse - 68 b.p.m., blood pressure was 158/88 mmHg, respiratory rate – 14, and body temperature was 38,1*C. The laboratory tests showed increased pro-inflammatory markers. Anterior and lateral chest roentgenography was made and a non-homogenic consolidation in the left lung radix was observed for the first time.  The treatment with antibiotics was prescribed. There were no signs of other organ system damage. Furthermore, a computed tomography scan revealed alterations in the posterior mediastinum (Figure 1). A tumour measuring up to 5.5 cm in its longest dimension, with contrast accumulation, was found, extending from the second to the fourth thoracic vertebra and situated near the posterior wall of the oesophagus.

The patient had primary (essential) hypertension; her BMI was 28,87 kg/m2. Her arterial hypertension was well-controlled with nebivolol and zofenopril. There were no allergies to medications. There was no gastritis, duodenitis with ulcers, osteopenia or osteoporosis in her prior medical history that might be related to hyperparathyroidism.

The patient was transmitted to the thoracic surgery department for further investigation of the tumour in the posterior mediastinum. Since there were no complaints or symptoms related to hyperparathyroidism, no laboratory tests were performed. She had no signs of choking; the oesophagus mucous membrane was normal but after esophagogastroduodenoscopy, the upper third oesophagus compression was seen. 

We agree that our case was confusing. The CT scan was made after the detection of unclear findings in her roentgenography. Routinely we start with plain lateral and anterior roentgenography. 

  1. We do not know the patient's laboratory values ​​at the time of diagnosis. Were serum calcium, PTH, etc. values ​​really normal? Perhaps she was receiving calcium-lowering medical treatment (such as cinacalcet). Provide a table with the patient's laboratory values ​​at the time of diagnosis (such as PTH, calcium, phosphorus, albumin, vitamin D, creatine, urine calcium level, etc.). Also, did she have osteoporosis or nephrolithiasis?

We agree that the work-up process could have been better. The patient was not taking any calcium-lowering medication. She was completely healthy and had only arterial hypertension in her medical history which was well-controlled with nebivolol and zofenopril. After discovering the tumour in the posterior mediastinum, we should have performed biochemical tests to see what type of tumour it is. This would have been very helpful for the pre-operative period and planning operation. The Patient had no pancreatitis or nephrolithiasis in her medical history. Her lab work-up is added to the paper.

  1. Although it is a very rare condition, such a mass in the posterior mediastinum could also be a pheochromocytoma or paraganglioma. I would definitely consider checking the catecholamine levels. In this way, I would have had the chance to prepare before surgery and would not have put the patient at risk. Have you considered checking the catecholamines? (To prevent a possible pheochromocytoma crisis.).

No, we did not think about paragangliomas or pheochromocytomas. Their behaviour is different, and the patient had no symptoms of these types of tumours: palpitations, headaches, episodic sweating. Her blood pressure was not as high as it would be in having pheochromocytoma and it was well-controlled by medication (usually it is very hard to control pheochromocytoma-caused high blood pressure with medication).

  1. Considering the possibility of seeding the tumor, I would not consider performing a biopsy on such a large mass in such a localization. I would consider performing surgery directly. What do you think about this?

We partly agree with you.  Although there was a possibility of performing a tumour excision without a biopsy to avoid tumour seeding, histological tumour type confirmation was helpful in differential diagnosis. Tumours such as lymphomas can be treated using medication and do not require surgical intervention. That is why we performed a transoesophageal endoscopic ultrasound-guided fine-needle aspiration biopsy.

  1. Please include the images of the histopathological examination (light microscopy imaging). Present which stains were performed histopathologically and which were positive and which were negative, in a tabular format.

The histopathological images are added and confirmed by a different pathologist. The second-opinion diagnosis was well-differentiated parathyroid adenoma (G2).

However, it is important to highlight that the differentiation of parathyroid adenoma and carcinoma of pathohistological imaging can be difficult since they look very similar to each other. Since there are no signs of recurrence after 3 years of follow-up, the tumour is expected to be benign.

Reviewer 2 Report

Comments and Suggestions for Authors

Thank you for the opportunity to review the article entitled "Giant parathyroid adenoma of the posterior mediastinum".

I have read the article, which attracts attention - it is a description of an interesting clinical case of a patient with giant ectopic parathyroid adenoma in an unusual location - posterior mediastinum.

Unfortunately, the case description is limited, it includes the results of the imaging test and the perioperative course, without going into preoperative details. It lacks, among other things, information on the reasons for referring the patient for imaging diagnostics or the results of laboratory tests (if available, concerning calcium and phosphate metabolism, but also others), as well as medical history (such as comorbidities that may be related to hyperparathyroidism: gastritis and duodenitis with ulcers, osteopenia and osteoporosis, psychological problems, cardiovascular diseases, etc.).

The authors describe that after the procedure, no signs of hyperparathyroidism were observed postoperatively (which is not surprising, as we would rather expect these signs in the preoperative period) - however, there is no broader description of the tests performed, such as PTH, calcium, phosphorus, etc., confirming the significant lack of signs of hyperparathyroidism in the postoperative period.

A valuable addition to the work presentation would be to include the histological imaging of the removed tumor. Do the authors have any such material available for inclusion in the article?

The "Discussion" and "Conclusions" sections do not raise any objections, they fulfill their proper roles.

Despite high hopes, the work requires significant expansion to be accepted for publication.

Author Response

Dear Reviewer,
Thank you very much for your time and comments. We agree that there was a lack of additional information regarding our patient's laboratory follow-up results, medical history, etc. We are sending a new version of the article. We added the patient's medical history, laboratory work-up results and histopathological imaging. I hope it will meet your expectations.
Have a great week, and thank you for improving us.

Reviewer 3 Report

Comments and Suggestions for Authors   1. The article introduces a giant patients with ectopic parathyroid adenoma, but the discussion did not discuss the difference between the treatment brought by the so-called "giant" and the conventional treatment in the past. 2. The size of this tumor is relatively large, but the author did not mention the approximate size in the past. 3. It seems that this tumor was found through imaging examination. Does the patient have a series of special symptoms due to the excessive size of the tumor? 4. I think the author should attach a list of blood drawing data for each case for readers’ reference.  

Author Response

Dear Reviewer,
Thank you very much for your time and comments. We agree that there was a lack of additional information regarding our patient's laboratory follow-up results, medical history, etc. We are sending a new version of the article. We added the patient's medical history, laboratory work-up results and histopathological imaging. I hope it will meet your expectations.

Does the patient have a series of special symptoms due to the excessive size of the tumor?
Answer: Unfortunately no. The patient had no complaints (signs of choking); the oesophageal mucosa was normal, but after esophagogastroduodenoscopy, the upper third oesophagus compression was seen. 

Have a great week, and thank you for improving us.

Round 2

Reviewer 2 Report

Comments and Suggestions for Authors

Thank you for the opportunity to review the second article entitled "Giant parathyroid adenoma of the posterior mediastinum".

The authors have responded to the comments from the previous review. Clinical information regarding the reasons for hospitalization and the clinical picture of the patient before, during, and after hospitalization has been supplemented. The article has been supplemented with histopathological data. All this has improved the quality of the manuscript.

Currently, the only comment I would like to make is the lack of reference values for the tested parameters given in Table 2 - I suggest the authors add them in the appropriate place in the table.

After this modification, I recommend the article for acceptance - until then, I suggest a minor revision.

Author Response

Dear Reviewer,

we added reference values used in Vilnius University Hospital Santaros Clinics laboratory in Table 2.
I hope that the modification will be sufficient. 

Thank you for helping us to improve our paper. 

Reviewer 3 Report

Comments and Suggestions for Authors

I have no more comments 

Author Response

Dear Reviewer,

Thank you very much for your response. Is there any information we should provide? Would you consider accepting our paper?

Have a great day!